# Quality and Safety of Proximity Care Centered on the Person and Their Home: A Systematic Review Protocol

**DOI:** 10.3390/ijerph20054504

**Published:** 2023-03-03

**Authors:** Carlos Martins, Ana Escoval, Manuel Lopes, Susana Mendonça, César Fonseca

**Affiliations:** 1Cova da Beira Group of Health Centers, 6200-034 Covilhã, Portugal; 2Comprehensive Health Research Center, University of Évora (CHRC-UE), 7000-811 Évora, Portugal; 3Department of Nursing, University of Évora, 7000-811 Évora, Portugal

**Keywords:** quality, safety, home health care, patient safety, home nursing

## Abstract

The quality and safety of health care is a priority, a requirement and a demand of health organizations and social institutions with concrete purposes of progressively providing people with a higher level of health and well-being. It is in the development of this path that home care currently represents an area of gradual investment and where health care services and the scientific community have shown interest in building circuits and instruments that can respond to needs. It is essencial that care must be centered and in close proximity to the person and their family, their context. On the other hand, in Portugal, there are already quality and safety models for the institutionalization context however it is non-existent for home care. In this sense, our objective is to identify, through a systematic review of the literature, particularly from the last 5 years, areas of quality and safety in home care.

## 1. Introduction

In recent years, changes in the demographic profile and the emergence of chronic diseases have translated into new “health needs and an increase in the complexity of health problems, associated, among others, with population aging and multiple morbidity and dependence, but also to a more acute awareness of access to health as a right” [1]. These particularities and complexities require that care be centered on the person, that is, that it respects and responds to their preferences, needs and individual values, thus ensuring that they guide all clinical decisions [2]; guarantee the integration of care, that is, that the person has access to the type and intensity of care that he effectively needs in the right place and in a timely manner; and the continuity of care, that is, that care is adequate for the transitions between the various levels of care [3].

Home care is defined as “care that aims to satisfy the social and health needs of people in their own homes” [1], these can respond to current challenges arising from changes in epidemiological and demographic profiles and it is pertinent to reflect on and rethink local health intervention strategies in a constant paradigm shift at the scientific, technical and organizational level, centered on the person and their context [4].

In order to respond to this complexity, health services must adapt and rethink their intervention with different institutions and community resources in an articulated way in an approach that goes beyond and across disciplines from a transdisciplinary perspective.

Transdisciplinarity is a theory of knowledge, establishing a dialogue between different areas of knowledge and understanding of processes in a new attitude towards the ability of understanding and responding to the multidimensionality of human beings and the world. Therefore, it is a pluralistic focus on knowledge that aims to achieve the unification of knowledge through articulation between the many faces of understanding the world. This implies the union of the most varied disciplines so that a broader exercise of human cognition becomes feasible [5].

Thus, considering that home care can respond to current challenges, it is pertinent to reflect and rethink local health intervention strategies in constant paradigm shifts at the scientific, technical and organizational level, centered on the person and their context [4]. 

This care delivery strategy includes interaction, integration, continuity, and closeness at home to achieve the person’s health and well-being and to guarantee high levels of quality.

In this perspective of care at home, in which the health professional travels to carry out specific interventions, the need for the involvement of the patient and the family in the whole process emerges as a way of guaranteeing care in the absence of professionals.

This results in an increase in the complexity of the team, namely in terms of quality and safety and, consequently, the need to train both as a way of guaranteeing those factors.

There is also a need to redefine the care process, not only by increasing its actors but also by changing the context to the home of each patient. In fact, the context does not just change, it changes according to each patient (Figure 1).

There is also a change in the relational model, insofar as it takes place in the context of people’s homes but also in the context of a care process that, obligatorily, integrates the patient and the family caregiver as team partners and simultaneously as recipients of care.

Thus, the home care process refers to having the sick person and the family/caregiver as the focus of care, despite the fact that the latter are also care partners; emphasizes the Individual Care Plan (ICP) that allows the sharing of information and communication between all stakeholders and expresses the centrality in the person; is redefined and provided in a coordinated manner, achieving this with the creation of tools and procedures for organization, intervention and decision-making [1].

Furthermore, from the perspective of Sousa (2006), with the organization and articulation of the work of professionals involved and the establishment of protocols and the adoption of guidelines and norms of clinical guidance, in order to reduce variability and increase informed decision-making based on the best available evidence, the “current paradigm”, prevents its fragmentation and the obtaining of unsatisfactory results, and also is a guarantee of its quality and safety [6].

However, these exciting times in health care, spurred on by the incredible advances in modern health care, make the shortcomings associated with delivering that care even more visible. Healthcare is a major source of preventable harm and patients are exposed to wide and unexplained variations in the quality of care they receive.

To circumvent these variations, it is essential to gather high-quality evidence-based to understand health care’s effectiveness, costs and risks. The lack of a scientific approach can lead to results that are absolutely opposite to those intended by improvement initiatives [7]. Hence the need to redefine the care process, not only by increasing its actors, but also by changing the context to the home of each patient. In fact, the context does not just change, it changes according to each patient. In addition, the relational model takes place in people’s homes, obligatorily, integrates the patient and the family caregiver as team partners, and simultaneously, as care recipients.

All of these elements call into question the classic trilogy of quality and safety-structure, process and results [8]—and therefore force us to rethink the trilogy because it is unquestionable that it is necessary to guarantee the quality and safety of home care.

Donabedian’s Quality Model integrated the notion of structure, process and outcome indicators into systems’ theory, thus adapting them to the health sector. The structure corresponds to the way the organization presents itself in relation to resources, norms, routines and the system of values and expectations, these characteristics being necessary for the care process. The process is related to how healthcare is being provided to the patient, according to established technical and scientific standards. The result, in turn, corresponds to the consequences of activities carried out at home or by the professionals involved [9,10,11].

In this sequence, the perspective of the American Institute of Medicine (IOM) in 1990, defines quality as “the extent to which health services provided to individuals and populations increase the probability of obtaining the desired health outcomes and are consistent with the current professional knowledge” [12], and extends this concept to the general population and to the consistency of results in the organization.

Regarding the concept of evaluating the quality of care, the WHO (2020) defined the dimensions of the IOM (2001) as being more consensual: efficiency, effectiveness, equity, person-centricity, safety and on-time/opportunity [12].

Highlighting the definitions of the WHO (2020, p. 13) for the recognition of quality health services worldwide, these must be:○Effective: delivering evidence-based health care to all those who need it;○Insurance: avoiding harming the people for whom care is intended;○People-centered: providing care that responds to the preferences, needs and values of individuals.

In addition, for the benefits of quality health care to be real, including home care, it will have to be:○Timely: reducing waiting times and sometimes harmful delays for both recipients and providers of care;○Equitable: providing care that does not vary in quality due to age, sex, gender, race, ethnicity, geographic location, religion, socioeconomic status, language issues or political affiliation;○Integrated: delivering care that is coordinated at all levels and by all providers and provides the full range of health services across the life course;○Efficient: maximizing the benefit of available resources and avoiding waste.

The WHO (2020) also mentions that there are many definitions of quality which may have a broad enough interpretation to define the perception of quality at a national, regional or health unit level [12].

However, in a systematic review on the quality of home care in Arab countries carried out by Al Anazi et al. (2020) with the aim of examining the quality of these services, the authors found that of all the studies analyzed, they were only evaluated according to three of the six quality indicators of the IOM: safety, efficiency and effectiveness [13].

These authors concluded that their review shows a clear gap in the literature on the quality of home care in Arab countries, emphasizing the need for more studies, especially quality studies on timely, equitable and patient-centered care in home health settings [13]. Furthermore, in Portugal, the existing models of quality and safety in the care delivery process were created and approved for areas of health care other than home care [1]. Thus, it becomes imperative to develop a care quality model at the level of services and home care in Primary Health Care in continental Portugal that considers the great variability in terms of the structure and actors in the care process; a relational model in a context of care that we do not fully control and that is always different.

Despite the specificities and particularities of home care, we intend to consider the perspective of the Donabedian triad, considering that the structure with its variability, the patient’s home and the process in which multiple caregivers participate may constitute a severe limitation to the quality and safety of care, with the analysis and evaluation of the results taking into account the previous considerations [1].

In view of the above, we consider this study we propose to develop to be extremely relevant due to the current lack of knowledge in this specific field of service and care, due to the importance that quality and patient safety has assumed in recent years, and also due to the growing concern on the part of political decision-makers, professionals and also patients and their families and/or caregivers.

In the research carried out as state of the art, focusing on the international literature in the last 5 years of scientific articles from the database on the EBSCO host platform, which integrates CINAHL Complete, MEDLINE Complete, Nursing & Allied Health Collection: Comprehensive, the Cochrane Central Register of Controlled Trials, the Cochrane Database of Systematic Reviews and MedicLatina, published in Portuguese, English, French and Spanish on the quality and safety of care, indicators and improvement of quality of services and home health care, person-centered care, and the professional–patient relationship, eight were screened as we considered them to have the greatest contribution to this theme.

In this context, the articles resulted in two dimensions. In the first dimension, the perspective examined was the one that the sick person and caregivers have in relation to home care, and where the studies were carried out by: LaFave et al. (2021); Sanerma et al. (2020); Bolenius et al. (2017) in Sweden; Róin (2018) in the Faroe Islands; Dostálová et al. (2022) in the Czech Republic; and Oosterveld-Vlug et al. (2019) [14,15,16,17,18,19]. In the second dimension, the perspective examined was that of professionals on quality and safety in home care, as carried out by Olsen (2021) in Norway; and Al Anazi et al. (2020) in Arab countries [13,20].

In the research carried out, we had some difficulty in obtaining studies on the quality and safety of home care; despite this constraint, the results found and selected highlight multiple quality indicators. Of these, those that fit the six dimensions of the IOM stand out, and also the tranquility of accessing care without leaving home, proximity and access to the health team were valued; there was an emphasis on the competence of professionals, availability, proactivity, serenity and kindness. These attributes are considered relevant to the provision of care and the assessment of quality and contribute to the continuous improvement of the quality of care at home.

The results also show more specific aspects in relation to home care, such as people’s lack of involvement in home care, which could affect their quality of life; the importance of knowing and understanding the path of the patient; the need to understand the role of home care providers; the importance of implementing quality checklists; as well as the need for more studies to be carried out in this area of home care.

Our review aims to deepen in a more systematic way the components (or areas) that may be involved in the quality and safety of care provided at home.

With this protocol, we intend to specify the conditions and criteria for carrying out this review and with its fulfilment guarantee the rigor, clarity and quality of the process. To this end, we always involved at least two reviewers in the multiple stages of article identification, in the search in different databases and in the selection of studies. As a result of this systematic review, we intend to contribute to scientific knowledge on the subject, identify guidelines for practice based on scientific evidence and inform possible guidelines for future research, namely, to contribute to the adoption of a quality and safety model for home care.

### 1.1. Aim

To identify through a systematic literature review, particularly the last 5 years, areas and criteria for quality and safety in home care.

### 1.2. To Review Questions

What is the state of the art in areas of quality and safety in home care?

What areas and criteria should exist for the construction of a health care model that guarantees the quality and safety of patients at home?

## 2. Materials and Methods

The protocol was developed in October 2022 in accordance with the Preferred Reporting Items for Systematic Reviews and Meta-Analyses (PRISMA) [21] and registered in December 2022 in the International Prospective Register of Systematic Reviews (PROSPERO) with the registration number: CRD42022380989.

Considering that the scope of this study was wide, we chose to include all types of primary quantitative or qualitative empirical studies, including cross-sectional, longitudinal, observational or experimental studies in this review.

### 2.1. Eligibility Criteria

#### 2.1.1. Population

The inclusion criteria was all studies written in Portuguese or English, published after 2017 in order to obtain scientific knowledge as up to date as possible. All secondary studies (such as reviews) and all those that did not present an abstract or full article were excluded.

#### 2.1.2. Intervention

The review included studies on areas of quality and safety in home care, in different geographic areas, communities, cultures or specific environments, with different work methodologies and organization of health and social services.

#### 2.1.3. Comparison

In this review, studies with or without a comparison group were included.

#### 2.1.4. Primary Result

The main objective was to identify and summarize areas of quality and safety in in-home care and then seek to identify the most desirable criteria for quality and safety levels of in-home care. The data must be qualitative or quantitative in nature, synthesized from primary empirical qualitative and quantitative studies: cross-sectional, longitudinal, observational or experimental.

#### 2.1.5. Study Design

The review included primary empirical qualitative or quantitative studies: cross-sectional, longitudinal, observational or experimental.

#### 2.1.6. Context

The review included all studies related to the areas of quality and safety in home care, regardless of the conjuncture of geographic areas, communities, cultures or specific environments, such as work methodologies and the organization of health services and different social institutions.

### 2.2. Search Strategy

#### 2.2.1. Data Sources

As a strategy, a broad bibliographic search in the databases was carried out: EBSCOhost Research Databases (CINAHL (Plus with Full Text), MEDLINE (Plus with Full Text; Psychology and Behavioral Sciences Collection)).

#### 2.2.2. Search Terms

The research combined the key concepts of the research question with the terms: ((“Home care services”) OR (“Housing”) OR (“Resistant homes”) OR (“Home nursing”) OR (“Home care”)) AND ((“Patient safety”) OR (“Patient safety indicators”) OR (“Safety”) AND (“Quality of service”) OR (“Quality healthcare”) OR (“Quality Indicators”) OR (“Quality”) AND (“IT Quality”)).

The strategy was adapted according to each database and restricted to studies in the time period between January 2017 and November 2022 in Portuguese or English.

### 2.3. Data Collection and Analysis

#### 2.3.1. Selection of Studies

Search results in each database were exported to Mendeley and duplicate studies were removed. In order to minimize bias, two reviewers independently assessed the inclusion of studies based on reading their titles and abstracts, excluding those that did not meet the inclusion criteria for this review. In case of divergence or doubts, a third reviewer was consulted. Subsequently, the full texts were evaluated and the PRISMA flowchart [21] was used to highlight the identified studies, that is, those that were screened and/or included.

#### 2.3.2. Data Extraction

Data were collected by at least two people, one of whom extracted data and another person verified the extracted data on the international literature search platform for scientific articles from the database on the EBSCOhost Research Databases platform that integrates CINAHL (Plus with Full Text) and MEDLINE (Plus with Full Text; Psychology and Behavioral Sciences Collection).

A minimum of five studies and a maximum of eight were selected based on the defined criteria. In the first phase of data extraction, a descriptive evaluation of each study was carried out using the extraction instrument designed to collect information according to the research question, being able to obtain other information, namely: the country where it was carried out, authors, year, study objective, sample, methods, results and conclusions. As previously mentioned, the extracted data were independently evaluated by two reviewers and any uncertainty or disagreement was resolved by a third party.

#### 2.3.3. Quality Assessment

As this is a review of qualitative and quantitative studies, with regard to the latter, the quality assessment tools were those of the Joanna Briggs Institute (JBI) to assist in the assessment of reliability, relevance and results of published articles [22], carried out by two reviewers independently, with disagreements with the assessment resolved by a third reviewer. 

#### 2.3.4. Data Synthesis Strategy

After the inclusion of the studies, the methodological evaluation was carried out, presenting their characteristics and finally elaborating the discussion and conclusion of the review. The synthesis and analysis of the results were of a narrative nature; structured to answer the presented research question.

In order to facilitate the analysis and discussion of the results, a data summary table was built. It intended to group and synthesize the available data of each study related to the answer and research question and grouped by the characteristics of the included studies. The country, author, year, type of study, objectives, sample, methods, evaluation instruments, interventions, results, conclusions and relevant comments were noted; a data compilation scheme was also made.

Tables, graphs and/or figures were prepared to present the results of the synthesized data and were arranged in the same way as the syntheses, described in the narrative text to facilitate the comparison and analysis of the findings of each included study.

In order to improve the presentation of data, all team members participated in this process. The involvement of patients and/or the public in the process of developing this review was not foreseen.

#### 2.3.5. Assessment of the Quality of the Evidence Produced by the Review

The GRADE (Grade of Recommendations Assessment, Development and Evaluation) protocol was used, which provided information on the presence of biases, inaccuracies or inconsistencies in results, among other important factors, to assess the quality of the study’s evidence [23,24]. 

Taking into account that the scope of this study was very specific and still understudied, we chose to include primary empirical quantitative studies, cross-sectional, longitudinal, observational or experimental studies in this review.

#### 2.3.6. Study Limitations and Concerns

Considering that only studies in English and Portuguese were included in the proposed research, this may lead to a language bias. Similarly, this study may be limited by the fact that only studies from the last 5 years were included, and secondary studies and those that do not present an abstract or complete article were also excluded.

The breadth of this study and the possible variability of contents is also a concern, as this was an international survey; therefore, different home care priorities may be found that change depending on the countries and/or regions.

## 3. Results

In this systematic review, four electronic databases were systematically searched, CINAHL Plus with Full Text, MedicLatina, MEDLINE with Full Text and the Psychology and Behavioral Sciences Collection, to identify eligible studies published in Portuguese or English, from January 2017 to November 2022. Following the PRISMA flowchart [21], 183 articles were screened, 44 duplicates were removed and 113 were excluded by title. Eleven were evaluated for eligibility and five were excluded due to the complete reading of the text, thus the six that met the inclusion criteria were selected.

## 4. Conclusions

This systematic review is the first phase of a study that will be carried out using a mixed methodology and developed in three stages: the first constituted of this systematic review to understand the state of the art and identify areas and criteria for the quality and safety of a care household. This stage of the study is considered the basis of the study, as it is through its elaboration that all relevant scientific evidence will be identified, which can support and guide the remaining investigation. The concern is to obtain a variability in the content and different forms of organization and priorities depending on the multiple countries and/or regions, whose diversity may contribute to enriching the knowledge and the obtaining of different perspectives, and to seek to contextualize the areas of quality and safety in home care and identify its main dimensions.

The second stage will aim to carry out a diagnosis of the current development of quality and safety methodologies in home care in Primary Health Care in continental Portugal, using a questionnaire. Stage three will consist of a qualitative study, using focus groups with the sample under study made up of stakeholders from the areas of program planning and coordination, teaching and home care, seeking to define a proposal for contributions on quality criteria and safety levels to be integrated into a quality and safety model for home care.

With the development of the study, the results are expected to contribute to the increase in scientific knowledge on the subject and it is intended to contribute to the adoption of a quality and safety model for home care.

## Figures and Tables

**Figure 1 ijerph-20-04504-f001:**
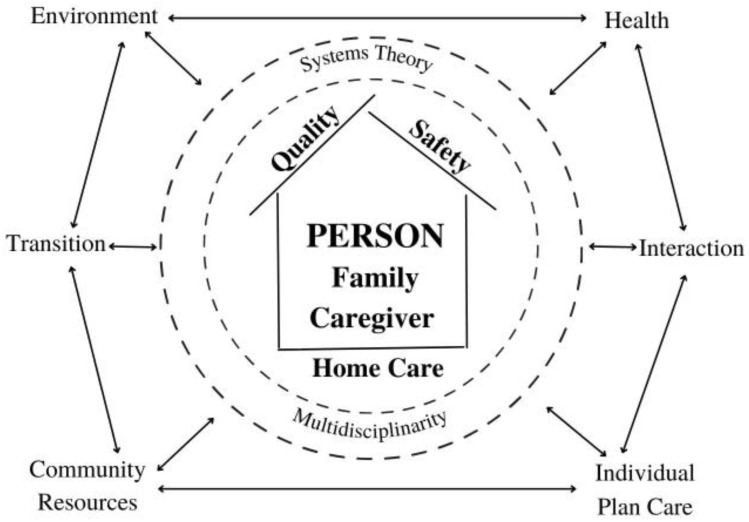
Patient Quality and Safety in Home Care.

## Data Availability

Not applicable.

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
