# Peer review of "Quality and Safety of Proximity Care Centered on the Person and Their Home: A Systematic Review Protocol"

_ijerph, 2023, doi:10.3390/ijerph20054504_

Round 1
Reviewer 1 Report
I am very happy to be able to contribute to the evaluation of this study, because home care is the future and we have to be prepared to offer quality care. Congratulations to the authors for the theme and here are some contributions and reflections for the authors.
1. Introduction
- Presents the contextualization of the theme in a logical sequence.
- I suggest explaining the knowledge gap in the scientific literature on the quality and safety of care at home, because as the authors pointed out, quality and safety models for the context at home in Portugal are non-existent, however, it is important that this gap be addressed evaluated in the literature in order to strengthen the need and relevance of this study. There are already studies with the application of the Donabedian model for Primary Health Care, however, I am not sure if these studies are specifically aimed at the home. To check.
- After presenting the knowledge gap, the authors need to justify the relevance of the study.
- In the 8th paragraph of the introduction “In Portugal, the existing models of quality and safety in the care delivery process were created and approved for areas of health care other than home care” – add references to this information.
- When working with the Donabedian triad, it is important to make clear what will be assessed, whether it is care specifically or the home care service (in its broadest form).
2. Method
- In the item Eligibility criteria, Population, I suggest justifying the use of studies after 2017.
3. Limitations
- it is important to highlight the limitations of the study.
Author Response
ThThanks for the suggestion
Point 1: Introduction: I suggest explaining the knowledge gap in the scientific literature on the quality and safety of care at home because as the authors pointed out, quality and safety models for the context at home in Portugal are non-existent, however, it is important that this gap be addressed evaluated in the literature in order to strengthen the need and relevance of this study. There are already studies with the application of the Donabedian model for Primary Health Care, however, I am not sure if these studies are specifically aimed at the home. To check.
- After presenting the knowledge gap, the authors need to justify the relevance of the study.
- In the 8th paragraph of the introduction “In Portugal, the existing models of quality and safety in the care delivery process were created and approved for areas of health care other than home care” – add references to this information.
- When working with the Donabedian triad, it is important to make clear what will be assessed, whether it is care specifically or the home care service (in its broadest form).
Response Point 1:
In response to the first suggestion, we have added a text that reinforces the importance of research in this area, which provides a framework for the theme, shows the relevance of the study, and highlights the reality in Portugal, taking into consideration the Donabedian Model. The figure also aims to clarify the components that we consider essential to a future model of care at home.
This study is essential to find factors that may influence the improvement of care in the home, for this it is essential to gather high-quality evidence-based evidence that determines its effectiveness, costs, and risks. In order to respond to the complexity of home care, it is necessary to know the factors to build a model adapted to home care. Because Portugal doesn’t have a model adapted to home care in primary healthcare.
In order to respond to this complexity, health services must adapt and rethink their intervention with different institutions and community resources in an articulated way, in an approach that goes beyond and across disciplines, in a transdisciplinary perspective.
Transdisciplinarity, being a theory of knowledge, establishing dialogue between different areas of knowledge and understanding of processes, in a new attitude, towards the ability to understand and respond to the multidimensionality of human beings and the world. Therefore, it is a pluralistic focus on knowledge that aims to achieve the unification of knowledge through articulation between the many faces of understanding the world. Implying the union of the most varied disciplines so that a broader exercise of human cognition becomes feasible [5].
Thus, considering that home care can respond to current challenges, it is pertinent to reflect and rethink about local health intervention strategies in constant paradigm shifts at the scientific, technical and organizational level, centred on the person and their context [4]. In a logic of care of interaction, integration, continuity and proximity, preferably at home, and in the results in health and well-being of the person and community, guaranteeing high levels of quality.
In this perspective of care at home, in which the health professional travels to carry out specific interventions, the need for the involvement of the patient and the family in the whole process emerges as a way of guaranteeing care in the absence of professionals.
This results in an increase in the complexity of the team, namely in terms of quality and safety and, consequently, the need to train both as a way of guaranteeing them.
There is also a need to redefine the care process, not only by increasing its actors, but also by changing the context: the home of each patient. In fact, the context does not just change, it changes according to each patient.
There is also a change in the relational model, insofar as it takes place in the context of people's homes, but also in the context of a care process that, obligatorily, integrates the patient and the family caregiver as team partners and simultaneously as recipients of care.
Thus, the home care process refers to having the sick person and the family/caregiver as the focus of care, despite the fact that the latter are also care partners; emphasizes the Individual Care Plan (ICP) that allows the sharing of information and communication between all stakeholders and expresses the centrality in the person; be redefined and provided in a coordinated manner, achieving with the creation of tools and procedures for organization, intervention and decision-making [1].
And also, from the perspective of Sousa (2006), with the organization and articulation of the work of the professionals involved, with the establishment of protocols and the adoption of guidelines, norms of clinical guidance, in order to reduce variability and increase informed decision-making based on the best available evidence, the “current paradigm”, preventing its fragmentation and obtaining unsatisfactory results, and also as a guarantee of its quality and safety [6].
Figure 1 - Patient Quality and Safety in Home Care
But these exciting times in healthcare, spurred on by the incredible advances in modern healthcare, make the shortcomings associated with delivering that care even more visible. Healthcare is a major source of preventable harm and patients are exposed to wide and unexplained variations in the quality of care they receive.
To circumvent these variations, it is essential to gather high-quality evidence-based evidence that determines its effectiveness, costs, and risks. The lack of a scientific approach can lead to results that are absolutely opposite to those intended by improvement initiatives [7].
Also highlighting the WHO (2020, p13) that for the recognition of quality health services worldwide, these must be:
- “Effective: delivering evidence-based health care to all those who need it.
- Insurance: avoiding harming the people for whom care is intended.
- People-centred: providing care that responds to the preferences, needs and values ​​of individuals.”
In addition, for the benefits of quality health care to be real, including home care, it will have to be:
- Timely: reducing waiting times and sometimes harmful delays for both recipients and providers of care.
- Equitable: providing care that does not vary in quality due to age, sex, gender, race, ethnicity, geographic location, religion, socioeconomic status, language issues or political affiliation.
- Integrated: delivering care that is coordinated at all levels and by all providers and provides the full range of health services across the life course.
- Efficient: maximizing the benefit of available resources and avoiding waste”.
The WHO (2020) also mentions that there are many definitions of quality, which may have a broad enough interpretation to define the perception of quality at national, regional or health unit level [12].
However, in a systematic review on the quality of home care in Arab countries, carried out by Al Anazi et al. (2020), with the aim of examining the quality of these services, the authors found that of all the studies analyzed, they were only evaluated according to three indicators, namely, safety, efficiency and effectiveness, of the six quality indicators of the IOM [13].
These authors concluded that their review shows a clear gap in the literature on the quality of home care in Arab countries, emphasizing the need for more studies, especially quality studies on timely, equitable and patient-centred care in home health settings. [13]. Also, in Portugal, the existing models of quality and safety in the care delivery process were created and approved for areas of health care other than home care [1]. Thus, it becomes imperative to develop a care quality model at the level of services and home care in Primary Health Care (CSP) in Continental Portugal that considers the great variability in terms of structure and actors in the care process; a relational model in a context of care that we do not fully control and that is always different.
Despite the specificities and particularities of home care, we intend to consider the perspective of the Donabedian triad, considering that the structure with its variability, the patient's home, and the process in which multiple caregivers participate may constitute a severe limitation to the quality and safety of care, with the analysis and evaluation of the results taking into account the previous considerations [1].
In view of the above, we consider the study that we propose to develop to be extremely relevant due to the lack of knowledge in this specific field of service and care, due to the importance that quality and patient safety has assumed in recent years, as well as due to the growing concern on the part of political decision-makers, professionals and also patients and their families and/or caregivers.
Point 2 - Method
- In the item Eligibility criteria, Population, I suggest justifying the use of studies after 2017.
Response Point 2: Inclusion criteria will be all studies written in Portuguese or English, published after 2017 in order to obtain scientific knowledge as up to date as possible. All secondary studies (such as reviews) and all those that do not present an abstract or full article will be excluded.
Point 3 – Limitations
- it is important to highlight the limitations of the study.
Response Point 3:
2.3.6. Study limitations and concerns
Considering that only studies in English and Portuguese will be included in the proposed research, this may lead to a language bias. Similarly, the study may be limited by the fact that only studies from the last 5 years are included, and secondary studies and those that do not present an abstract or complete article are also excluded.
The breadth of the study and the possible variability of contents is also a concern, as this being an international survey, different home care priorities may be found that change depending on the countries and/or regions.
"Please see the attachment"

Reviewer 2 Report
Dear authors, the topic of your paper is very relevant today and, in my humble opinion, more in the future.
I think that home care safety is just one of the various topics to be addressed in the transition to home care. Others items linked with safety are related to the digital ecosystem and the communication between healthcare professionals and patients. For this reason I suggest you to evaluate if other terms should be included in the research ("IT ecosystem", "IoT", "IoMT", "connected devices", together with cybersecurity topics).
I'm very curious about final results.
A very secondary note: at row 171 a " is missing before IT Quality".
Author Response
Point 1:
I think that home care safety is just one of the various topics to be addressed in the transition to home care. Others items linked with safety are related to the digital ecosystem and the communication between healthcare professionals and patients. For this reason I suggest you to evaluate if other terms should be included in the research ("IT ecosystem", "IoT", "IoMT", "connected devices", together with cybersecurity topics).
I'm very curious about final results.
A very secondary note: at row 171 a " is missing before IT Quality".
Response Point 1:
Thanks for the suggestion, we will take this into account when systematizing and organizing the information collected. Because it also seems essential to us in ensuring safety in health care, our ambition through the systematic review is to find components of the IT ecosystem that show how health professionals use them in their daily practice to ensure safety in the transition and continuity of care.
The research will combine the key concepts of the research question, with the terms: ((“Home care services”) OR (“Housing”) OR (“Resistant homes”) OR (“Home nursing”) OR (“Home care”)) AND ((“Patient safety”) OR (“Patient safety indicators”) OR (“Safety”) AND (“Quality of service”) OR (“Quality healthcare”) OR (“Quality Indicators”) OR (“Quality”) AND (“IT Quality”).
"Please see the attachment"

Reviewer 3 Report
Thank you for the opportunity to review this manuscript. This systematic review protocol describes a planned study to identify areas of quality and safety in home care. A dearth in the literature and significance of undertaking the study is clearly pointed out. The methods are robust and will address the protocol aim. Plans for the research following this review are indicated clearly. Furthermore, a clear theoretical basis underpinning the review is described—the introduction describes an impetus for re-evaluating homecare safety and current quality paradigms based on Donabedian.
My two concerns are as follows: 1) The study aim and corresponding search strategy is very broad and will likely produce a wide variety of results. This may prove difficult to synthesize results. While I recognize that a sub-set of results from 2017 are reported in the “Results” section, the variety of content more so than the quantity of results is the concern; and 2) This review seems to target the international literature. Home care priorities are different across countries/regions. Will the research contextualize the areas of quality and safety in home care to identify dimensions that may vary by setting?
Author Response
Point 1:
My two concerns are as follows: 1) The study aim and corresponding search strategy is very broad and will likely produce a wide variety of results. This may prove difficult to synthesize results. While I recognize that a sub-set of results from 2017 are reported in the “Results” section, the variety of content more so than the quantity of results is the concern; and 2) This review seems to target the international literature. Home care priorities are different across countries/regions. Will the research contextualize the areas of quality and safety in home care to identify dimensions that may vary by setting?
Thanks for the suggestion, we improved the objective, added a question and study limitations.
- Aim:
To identify through systematic literature review, particularly the last 5 years, areas and criteria for quality and safety in home care.
- To review Questions:
What is the state of the art in areas of quality and safety in-home care?
What areas and criteria should exist for the construction of a health care model that guarantees the quality and safety of patients at home?
2.3.6. Study limitations and concerns
Considering that only studies in English and Portuguese will be included in the proposed research, this may lead to a language bias. Similarly, the study may be limited by the fact that only studies from the last 5 years are included, and secondary studies and those that do not present an abstract or complete article are also excluded.
The breadth of the study and the possible variability of contents is also a concern, as this being an international survey, different home care priorities may be found that change depending on the countries and/or regions.
Response Point 2:
Thanks for the suggestions, it is true that the realities of countries and contexts are specific, however, our goal is to gather data that can guide us in building a model that fits our health care and Portuguese reality.
- Conclusions
This systematic review is the first phase of a study that will be carried out using a mixed methodology and developed in three stages: the first constituted by this systematic review to know the state of the art and identify the areas and criteria forquality and safety of care household. This stage of the study is considered the basis of the study, as it is through its elaboration that all relevant scientific evidence will be identified, which can support and guide the remaining investigation.
Despite the concern to obtain a variability of contents and different forms of organization and priorities depending on the multiple countries and/or regions, whose diversity may contribute to enriching knowledge and obtaining different perspectives, and to seek to contextualize the areas of quality and safety in home care and identify its main dimensions.
"Please see the attachment"

Round 2
Reviewer 3 Report
Thank you for the response to the reviewer comments. The revisions and responses sufficiently address my comments.